# ACAGT-007a, an ERK MAPK Signaling Modulator, in Combination with AKT Signaling Inhibition Induces Apoptosis in KRAS Mutant Pancreatic Cancer T3M4 and MIA-Pa-Ca-2 Cells

**DOI:** 10.3390/cells11040702

**Published:** 2022-02-17

**Authors:** Golam Iftakhar Khandakar, Ryosuke Satoh, Teruaki Takasaki, Kana Fujitani, Genzoh Tanabe, Kazuko Sakai, Kazuto Nishio, Reiko Sugiura

**Affiliations:** 1Laboratory of Molecular Pharmacogenomics, Department of Pharmaceutical Sciences, Faculty of Pharmacy, Kindai University, Osaka 577-8502, Japan; iftakharkhandakar@outlook.com (G.I.K.); satohr@phar.kindai.ac.jp (R.S.); takasaki@phar.kindai.ac.jp (T.T.); fujitani.kana@kindai.ac.jp (K.F.); 2Laboratory of Organic Chemistry, Department of Pharmacy, Faculty of Pharmacy, Kindai University, Osaka 577-8502, Japan; g-tanabe@phar.kindai.ac.jp; 3Department of Genome Biology, Kindai University School of Medicine, Osaka 589-8511, Japan; kasakai@med.kindai.ac.jp (K.S.); knishio@med.kindai.ac.jp (K.N.)

**Keywords:** pancreatic cancer, ACAGT-007a (GT-7), ERK MAPK, apoptosis, AKT signaling, KRAS

## Abstract

The mitogen-activated protein kinase (MAPK)/ERK and phosphatidylinositol-3 kinase (PI3K)/AKT pathways are dysregulated in various human cancers, including pancreatic ductal adenocarcinoma (PDAC), which has a very poor prognosis due to its lack of efficient therapies. We have previously identified ACAGT-007a (GT-7), an anti-cancer compound that kills ERK-active melanoma cells by inducing ERK-dependent apoptosis. Here, we investigated the apoptosis-inducing effect of GT-7 on three PDAC cell lines and its relevance with the MAPK/ERK and PI3K/AKT signaling pathways. GT-7 induced apoptosis in PDAC cells with different KRAS mutations (MIA-Pa-Ca-2 (KRAS G12C), T3M4 (KRAS Q61H), and PANC-1 (KRAS G12D)), being T3M4 most susceptible, followed by MIA-Pa-Ca-2, and PANC-1 was most resistant to apoptosis induction by GT-7. GT-7 stimulated ERK phosphorylation in the three PDAC cells, but only T3M4 displayed ERK-activation-dependent apoptosis. Furthermore, GT-7 induced a marked down-regulation of AKT phosphorylation after a transient peak in T3M4, whereas PANC-1 displayed the strongest and most sustained AKT activation, followed by MIA-Pa-Ca-2, suggesting that sustained AKT phosphorylation as a determinant for the resistance to GT-7-mediated apoptosis. Consistently, a PI3K inhibitor, Wortmannin, abolished AKT phosphorylation and enhanced GT-7-mediated apoptosis in T3M4 and MIA-Pa-Ca-2, but not in PANC-1, which showed residual AKT phosphorylation. This is the first report that ERK stimulation alone or in combination with AKT signaling inhibition can effectively induce apoptosis in PDAC and provides a rationale for a novel concurrent targeting of the PI3K/AKT and ERK pathways.

## 1. Introduction

Pancreatic adenocarcinoma (PDAC) is one of the most lethal malignancies with a poor prognosis and high mortality rate. PDAC is also notorious for its characteristic resistance to conventional chemotherapeutic regimens, including gemcitabine, while other available therapeutic options are still limited.

The RAS/RAF/MEK/extracellular-signal-regulated kinase (ERK) MAPK and the PI3K/AKT pathways are frequently dysregulated in various human cancers as a result of genetic alterations in their components or upstream activation of cell-surface receptors [1]. Especially, activating point mutations of the RAS family genes (HRAS, KRAS, and NRAS) are a hallmark in PDAC, occurring in 90–95% of cases [2,3]. Oncogenic KRAS drives downstream activation of RAF/MEK/ERK and PI3K/AKT signaling, which promotes survival, invasion, and migration of cancer cells [4]. These findings link mutations in the KRAS proto-oncogene to the development of pancreatic cancer and place the downstream ERK and AKT signaling pathways as potential therapeutic targets for the development of pharmacologic inhibitors for the treatment of PDAC [5]. However, despite recent advances in the understanding of its molecular biology, therapies targeting PDAC-associated molecular pathways, including RAF/MEK/ERK signaling inhibitors, have not provided satisfactory results [2], partly due to the rapid up-regulation of compensatory alternative pathways and feedback loops within tumor cells. Therefore, there is a desperate need for developing innovative therapeutic approaches to fight this deadly malignancy.

Recently, our chemical genetic screen has identified ACA-28, which is a novel compound with a property that induces ERK-dependent apoptosis [6]. ACA-28 and its lead derivative ACAGT-007a (GT-7; described as **2b** in the previous reports) selectively induce apoptosis in several ERK-active melanoma cell lines, including SK-MEL-28, expressing the oncogenic mutant BRAF (V600E), by further stimulating ERK phosphorylation levels [7]. Notably, recent studies have highlighted the pro-apoptotic functions of ERK1/2 kinases and various compounds, including betulinic acid, piperlongumine, and cisplatin, trigger cancer cell death by enhancing ERK1/2 signaling [8,9]. Although the list of the compounds which mediate ERK activation and apoptosis is expanding, the ERK-dependent apoptosis in pancreatic cancer by an anti-cancer compound has never been reported.

In this study, we demonstrated that GT-7 induced apoptosis by stimulating ERK activation in T3M4, a PDAC cell line harboring the KRAS Q61H mutant allele. We investigated three PDAC cell lines with different KRAS mutations (MIA-Pa-Ca-2 (KRAS G12C), T3M4 (KRAS Q61H), and PANC-1 (KRAS G12D)) because different KRAS mutants in PDAC can lead to distinct biological manifestations, including the differences in downstream signaling pathways and various phenotypes, including susceptibility to chemotherapy [10]. Importantly, GT-7 successfully induced ERK-dependent apoptosis in T3M4 cells but not in the other two PDAC cells. We further showed that the difference in the AKT phosphorylation levels upon GT-7-treatment in each PDAC cell line is a key determinant for the resistance to apoptosis induction by GT-7. Consistently, co-treatment with GT-7 and the PI3K/AKT signaling inhibitor Wortmannin induced a more pronounced apoptosis induction than did GT-7 alone. Our data showed the potential of the GT-7 combination therapy as a therapeutic approach for PDAC. The possible mechanisms of the GT-7-induced apoptosis induction in PDAC cell lines in relation to the relevant KRAS mutations and the downstream ERK and AKT signaling will be discussed.

## 2. Materials and Methods

### 2.1. Cell Culture

The human pancreatic ductal adenocarcinoma (PDAC) cell lines MIA-Pa-Ca-2, T3M4, and PANC-1 were purchased from the American Type Culture Collection (ATCC) (Manassas, VA, USA). Cells were cultured in Dulbecco’s modified Eagle’s medium (DMEM) (Nacalai Tesque, Kyoto, Japan), supplemented with 10% fetal bovine serum (BioWest, Nuaillé, Pays De La Loire, France), sodium pyruvate, phenol red, and L-glutamine. The cells were maintained in a humidified incubator containing 10% CO_2_ at 37 °C. Every second–third day, the medium was changed, and cells were subcultured with 0.25% Trypsin-EDTA (Gibco, Invitrogen, Carlsbad, CA, USA) at 37 °C when confluency reached 70–80%.

### 2.2. Chemicals and Reagents

ACAGT-007a (GT-7) was synthesized as described previously (described as **2b**) [7] and dissolved in DMSO. Honokiol (H669560) was purchased from Toronto Research Chemicals (Toronto, ON, Canada) and dissolved in DMSO. U0126 (U-6770) was purchased from LC Laboratories (Woburn, MA, USA) and dissolved in DMSO. Wortmannin (HY-10197) was purchased from MedChemExpress (Monmouth Junction, NJ, USA) and dissolved in DMSO. Perifosine (P-6522) was purchased from LC Laboratories and dissolved in water.

### 2.3. WST-8 Assay

The assay was carried out using the Cell Count Reagent SF (Nacalai tesque, Kyoto, Japan) according to the manufacturer’s instructions with small modifications. Briefly, 100 μL of mammalian cell suspension at a cell density of 5.0 × 10^4^ cells/mL was seeded in a 96–well plate (IWAKI, Shizuoka, Japan) and incubated for 24 h. In each medium, the compounds in solution were diluted 1:1000, and 100 μL of the diluted compounds were added to the cell culture. Cells treated with solvent (DMSO) were used as controls. After incubation for 48 h, 6 μL of the Cell Count Reagent SF and 40 μL of the medium were mixed and added to the plate, followed by incubation for an additional 3 h. Then, the absorbance at 450 nm was measured by a Sunrise microplate reader (Tecan, Männedorf, Switzerland). Absorbance at 600 nm was also measured as the reference.

### 2.4. Protein Extraction and Western Blot Analysis

One milliliter of the PDAC cells suspension at a cell density of 2.0 × 10^5^ cells/mL was seeded in a 6-well plate (IWAKI, Shizuoka, Japan) and incubated for 24 h. After 24 h incubation, the medium was aspirated. Then, the cells were treated with chemicals required for the respective experiments as described in each figure legend. Harvested cells were lysed as previously described [6]. The following primary antibodies were used: anti-GAPDH (14C10) Rabbit mAb (Cell Signaling Technology, Danvers, MA, USA, #2118), anti-Caspase-3 Antibody (Cell Signaling Technology, Danvers, MA, USA, #9662), anti-Phospho-p44/42 MAPK (Erk1/2) (Thr202/Tyr204) Antibody (Cell Signaling Technology, Danvers, MA, USA, #9101), anti-p44/42 MAPK (Erk1/2) Antibody (Cell Signaling Technology, Danvers, MA, USA, #9102), anti-Phospho-Akt (Ser473) (D9E) XP^®^ Rabbit mAb (Cell Signaling Technology, Danvers, MA, USA, #4060), anti-Akt (pan) (C67E7) Rabbit mAb (Cell Signaling Technology, Danvers, MA, USA #4691), and anti-PTEN (138G6) Rabbit mAb (Cell Signaling Technology, Danvers, MA, USA, #9559). As a secondary antibody, anti-rabbit (#7074) IgG HRP-linked antibody (Cell signaling Technology, Danvers, MA, USA) was used. The proteins were detected by Chemi-Lumi One Super (Nacalai Tesque, Kyoto, Japan) or ECL Select (Cytiva, Marlborough MA, USA). Relative intensities of all bands were quantified using MULTI GAUGE Ver. 3.2 software (Fujifilm, Tokyo, Japan).

### 2.5. Small Interfering RNA Transfection

For small interfering RNA (siRNA) experiments, a human PTEN siRNA (5′-CCACACGACGGGAAGACAAGUUCAU-3′, 5′-AUGAACUUGUCUUCCCGUCGUGUGG-3′) and a control siRNA (Stealth RNAi™ Negative Control Med GC Duplex) were purchased from Invitrogen USA (PTEN siRNA; VHS41286 and control siRNA; 12935300). One milliliter of 1.25 × 10^5^ cells/mL was seeded in a 6-well plate (IWAKI, Shizuoka, Japan). A total of 7.5 μL of Lipofectamine™ RNAiMAX Transfection Reagent (Invitrogen, Carlsbad, CA, USA; 13778150) was diluted in 125 μL of Opti-MEM^®^ Reduced Serum Media (Gibco, Invitrogen, Carlsbad, CA, USA; 31985062). Simultaneously, 2.5 μL of 20 μM siRNA was also diluted in 125 μL of Opti-MEM^®^ Reduced Serum Media. Then, diluted Lipofectamine™ and diluted siRNA were mixed (1:1 ratio) and incubated for 10 min at room temperature. Finally, the siRNA–lipid complex was added to the seeded cells, and the transfected cells were incubated for 48 h.

### 2.6. Microscopy

One milliliter of the PDAC cells suspension at a cell density of 2.0 × 10^5^ cells/mL was seeded in a 35 mm glass-bottom dish (Appendix A; D11130H, MATSUNAMI, Osaka, Japan) or a 6-well plate (Appendix A; IWAKI, Shizuoka, Japan), and incubated for 24 h. After the incubation, the compounds in solution were diluted at 1:2000 by medium, and 1 mL of the diluted compounds were added to the cell culture. After an additional 24 h incubation, cells were observed by phase-contrast microscopy (BZ-X700/710, KEYENCE, Osaka, Japan) with a 10× (Appendix A) or 4× (Appendix A) magnification.

### 2.7. Flow Cytometry (FCM) Analysis of Apoptotic Cell

One milliliter of the PDAC cells suspension at a cell density of 2.0 × 10^5^ cells/mL was seeded in a 6-well plate (IWAKI, Shizuoka, Japan) and incubated for 24 h. After 24 h incubation, the cells were treated with chemicals required for the respective experiments as described in each figure legend. Apoptotic cells were stained by an eBioscience™ Annexin V-FITC Apoptosis Detection Kit (eBioscience, San Diego, CA, USA). Chemical-treated cells were washed by 1 mL of PBS(–), then resuspended by 100 μL of Binding Buffer (1×). In total, 2.5 μL of Annexin V-FITC and 5 μL of Propidium Iodide (20 μg/mL) were added. After 10 min incubation at room temperature, cells were diluted by 500 μL of PBS(–) and analyzed using FACS Calibur and LSR-Fortessa flow cytometers (BD Biosciences, Franklin Lakes, NJ, USA) and FlowJo (BD Biosciences, Franklin Lakes, NJ, USA).

### 2.8. RNA Isolation and qRT-PCR

Total RNA was isolated from PDAC cells using RNeasy Mini Kit (Qiagen, Hilden, Germany) according to the manufacturer’s instructions. In total, 0.1 μg of total RNA was reverse transcribed using ReverTra Ace^TM^ qPCR RT Master Mix with gDNA Remover (TOYOBO, Osaka Japan) according to the manufacturer’s instructions. Finally, 500-fold diluted cDNA was amplified using LightCycler^®^ 480 SYBR Green I Master (Roche, Basel, Switzerland) and primer sets (Appendix A), and the amplicon was detected and analyzed by LightCycler480 System II (Roche, Basel, Switzerland).

### 2.9. Statistical Analysis

Each experiment was performed 3 times. Representative data of at least three individual experiments were shown in Figure 1B, Figure 2A,C, Figure 3A, Figure 4A,C,D and Appendix A. A Student’s *t*-test was used to examine the differences between the two conditions in Figure 1A, Figure 2B and Figure 4E. Repeated ANOVA was used for multiple comparisons in Figure 3B and Figure 4B. *p* values of less than 0.05 are judged statistically significant. Values are shown as means ± standard error of the mean (SEM).

### 2.10. Data Access

Gene expression and mutation data were downloaded from the Cancer Cell Line Encyclopedia database (https://sites.broadinstitute.org/ccle/, accessed on 29 August 2021) [11].

## 3. Results

### 3.1. ACAGT-007a (GT-7) Reduces Cell Viability and Induces Apoptosis in PDAC Cells

Three PDAC cell lines with different genetic backgrounds regarding KRAS mutations (MIA-Pa-Ca-2, T3M4, and PANC-1) were tested for sensitivity to GT-7 in vitro using a cell viability assay (Figure 1A). T3M4 cells harbor the KRAS Q61H mutation, which is found in about 5% of PDAC patients and the most prevalent mutation occurring at codon 61 of KRAS [12]. PANC-1 cells harbor the KRAS G12D mutation, which is most prevalent in PDAC. MIA-Pa-Ca-2 cells harbor the KRAS G12C mutation, which comprises nearly 4% of PDAC [13].

In total, 30 μM of GT-7 treatment for 48 h suppressed the viability of MIA-Pa-Ca-2 cells to 18%, and that of T3M4 cells to 28%. In contrast, PANC-1 cells viability kept above 50% with 30 μM of GT-7 as compared with the vehicle, which indicates a relative resistance against GT-7. IC_50_ values of GT-7 for these cell lines were determined 48 h after treatment (Figure 1A). Briefly, MIA-Pa-Ca-2 (IC_50_: 7.3 μM) displayed the most sensitive response, and PANC-1 (IC_50_: 31.0 μM) showed the least sensitivity to GT-7 treatment. T3M4 (IC_50_: 21.4 μM) cells displayed an intermediate sensitivity. We also compared the IC_50_ values of GT-7 in these cell lines with those of Honokiol, a natural compound previously established as a promising candidate for cancer prevention and/or treatment for pancreatic cancer [14]. As shown in Figure 1A, GT-7 was shown to exhibit superiority in killing PDAC cell lines compared with Honokiol, as Honokiol needs significantly higher concentrations to kill the three PDAC cell lines (Figure 1A). Moreover, 15 μM of Honokiol did not affect the viability of the three PDAC cell lines, whereas the same concentrations of GT-7 significantly inhibited the cell viability except for PANC-1. It should be noted that GT-7 effectively reduced the cell viability of MIA-Pa-Ca-2 cells, which have been reported to display malignant phenotypes among PDAC, including the EMT phenotype as well as chemoresistance [15].

To investigate whether the effects of GT-7 on the PDAC cell viability was because of the induction of cell apoptosis, the three PDAC cells were treated with 0, 10, and 30 µM GT-7 for 24 h and analyzed by flow cytometry (FCM) using Annexin V and propidium iodide (PI). Annexin V/PI staining of the three PDAC cells showed that the percentage of apoptotic PDAC cells increases in a GT-7-dose-dependent manner (Figure 1B). Importantly, at a lower concentration of 10 µM GT-7, the susceptibility to apoptosis among the three PDAC cells is more evident; T3M4 displayed the strongest susceptibility (77.1%), followed by MIA-Pa-Ca-2 (47.4%) (Figure 1B). Again, PANC-1 cells are relatively most resistant (32.2%). At a higher concentration of 30 µM GT-7, severer cell death in nearly all cells (except for PANC-1; 60.2%) were induced (Figure 1B).

### 3.2. ACAGT-007 Induced ERK-Dependent Apoptosis in T3M4, but Not in MIA-Pa-Ca-2 and PANC-1 Cells

To investigate the underlying mechanism of GT-7-mediated apoptosis in the three PDAC cell lines, we examined the effect of GT-7 on ERK signaling and its relevance with apoptosis. We conducted immunoblot analysis using the anti-ERK and anti-phospho-ERK antibodies. GT-7 stimulated ERK phosphorylation in all three PDAC cell lines in a dose-dependent manner (Figure 2A,B). The GT-7-induced ERK activation was detected within 2 h exposure of PDAC cells to GT-7. We also analyzed the apoptosis induction by GT-7 in the three PDAC cells by immunoblotting using anti-Caspase-3 antibodies. Relative quantification of Caspase-3 cleavage showed that significant apoptosis induction was detected in T3M4 and MIA-Pa-Ca-2 in a GT-7 dose-dependent manner (Figure 2B). In contrast, cleaved Caspase-3 was barely detectable in PANC-1 cells, again indicating that PANC-1 cells were most resistant to GT-7-induced apoptosis.

To investigate if the GT-7-induced apoptosis was mediated by stimulation of ERK phosphorylation in PDAC cells, the effect of the MEK inhibitor U0126 was analyzed. ERK phosphorylation levels both before and after the addition of GT-7 were significantly blocked by U0126 in three PDAC cell lines (Figure 2A,B). Notably, relative quantification of Caspase-3 cleavage showed that GT-7-mediated induction of caspase-3 cleavage was blocked by the MEK inhibitor U0126, only in T3M4, but not in the other cell lines (Figure 2A,B).

To evaluate the importance of the ERK activation in the GT-7-mediated cell death mechanisms, FCM analysis was also performed (Figure 2C). To analyze the susceptibility of the three PDAC cell lines to GT-7-mediated cell death, 10 μM GT-7 was used. Consistent with the data obtained in Figure 1B, T3M4 cells were most susceptible, followed by MIA-Pa-Ca-2, and PANC-1 was most resistant. Blocking the ERK phosphorylation by U0126 significantly reduced the cell death rate in T3M4 (from 79.5% to 38.6%), whereas only a slight reduction in the cell death rate was detected in MIA-Pa-Ca-2 (from 47.4% to 44.1%) (Figure 2C). PANC-1 cells were insensitive to U0126 (from 32.0% to 31.0%) (Figure 2C). These results suggest that GT-7-induced ERK stimulation is required for the apoptosis induction in T3M4 cells harboring the KRAS Q61H mutation, which reproduced the findings in ERK-active melanoma cell lines with different oncogenic mutations, including BRAF [16].

### 3.3. AKT Signaling Activation Correlates with the PDAC Cells’ Resistance to Apoptosis Induction by GT-7

We next investigated the effect of GT-7 treatment on both the AKT and ERK signaling, as various KRAS mutations aberrantly activate downstream effector signaling, including the ERK and AKT signaling pathways. We then investigated the time course of both ERK and AKT activation upon GT-7 treatment by using the anti-phospho-ERK and anti-phospho-AKT (Ser473) antibodies, respectively. ERK activation was detected within 2 h exposure of the cells to 30 μM GT-7, and sustained ERK activation at variable duration was observed in three PDAC cell lines (Figure 3A,B). PANC-1 cells maintained significant ERK activation for 4 h, while MIA-Pa-Ca-2 and T3M4 cells maintained sustained ERK activation even after 10 h incubation with GT-7 (*p* < 0.01). Thus, in terms of sustained ERK activation, one of the hallmarks of the ERK-dependent apoptosis, PANC-1 showed less favorable ERK signaling dynamics when treated with GT-7, as compared with MIA-Pa-Ca-2 and T3M4.

Transient activation of AKT signaling was detected in the three PDAC cell lines exposed to GT-7 for 2 h (*p* < 0.01) (Figure 3A,B). However, the three PDAC cells displayed distinct duration and magnitude of AKT activation in response to GT-7 treatment. PANC-1 cells displayed the strongest and most sustained AKT activation. In PANC-1 cells, GT-7 induced a marked activation (approximately three-fold of the basal level) of AKT upon 2 h incubation, and sustained and higher levels of AKT signaling activity were observed after prolonged exposure to GT-7 (Figure 3A,B). MIA-Pa-Ca-2 displayed an intermediate phenotype in terms of AKT activation levels exposed to GT-7 for longer than 4 h (Figure 3A,B). Notably, T3M4 displayed a marked down-regulation of AKT phosphorylation levels below the basal levels after a transient increase in the AKT phosphorylation 2 h after the GT-7 treatment (Figure 3A,B).

We also performed the time-course analysis of the apoptosis induction in these PDAC cells after GT-7 addition. Quantification of the cleaved Caspase-3 showed that significant apoptosis induction was detected in T3M4 and MIA-Pa-Ca-2 4 h after the addition of GT-7 (Figure A,B). Again, PANC-1 cells were resistant to the GT-7-mediated apoptosis (Figure 2A). Thus, stronger susceptibility to apoptosis induction by GT-7 in T3M4 as compared with MIA-Pa-Ca-2 is likely to be derived from a significant decrease in AKT phosphorylation after a transient peak in T3M4 cells. Phase-contrast images of the three PDAC cell lines treated with GT-7 during the time course reflect the cell death status detected using immunoblot against cleaved Caspase-3 (Appendix A).

The time-course analysis of the ERK and AKT activation status in comparison with the cleaved Caspase-3 suggested that ERK activation was commonly observed in the three PDAC cells with a less sustained duration in PANC-1 cells. Thus, sustained AKT signaling activation levels in each PDAC cell line correlate with the cells’ resistance to apoptosis induction by GT-7. PANC-1 displayed the most sustained AKT activation, followed by MIA-Pa-Ca-2. In particular, a marked down-regulation of AKT phosphorylation below basal levels was observed in T3M4, coincident with the apoptosis induction as detected by the Caspase-3 cleavage. Thus, activation of ERK combined with inhibition of AKT signaling below a certain level may be required to make cells susceptible to GT-7-mediated apoptosis.

### 3.4. AKT Inhibition by Wortmannin Enhanced the ACAGT-007a-Induced Apoptosis in MIA-Pa-Ca-2 and T3M4 but Not in PANC-1 Cells

To determine whether the differences in the dynamics of AKT activation had any relationship with the cells’ resistance to apoptosis induced by GT-7, we tested the effect of Wortmannin and Perifosine, well-established inhibitors of PI3K and AKT, respectively. Wortmannin significantly inhibited the AKT activity as evaluated by the phospho-AKT levels (Ser473) in MIA-Pa-Ca-2 and T3M4 cells both before and after the addition of GT-7 (Figure 4A,B). Notably, in PANC-1 cells Wortmannin significantly inhibited the phospho-AKT at the basal levels but failed to do so upon GT-7 treatment (Figure 4A,B). Similarly, Perifosine markedly diminished AKT phosphorylation levels both before and after the addition of GT-7 in MIA-Pa-Ca-2 and T3M4 cells (Figure 4A,B). However, in PANC-1 cells, Perifosine only inhibited AKT phosphorylation at the basal levels, and GT-7-induced AKT phosphorylation was resistant to Perifosine (Figure 4A,B). The quantification of cleaved Caspase-3 showed that a single treatment with Wortmannin or Perifosine did not induce apoptosis in any of the PDAC cell lines, indicating that inhibition of PI3K/AKT signaling alone is not sufficient to induce apoptosis in PDAC (Figure 4A,B).

Importantly, the apoptosis induced by GT-7 was markedly enhanced by Wortmannin treatment in MIA-Pa-Ca-2 and T3M4 cells but not in PANC-1 cells (Figure 4A). The resistance of PANC-1 cells against Wortmannin in terms of enhancement of GT-7-mediated apoptosis may be explained by the residual AKT phosphorylation levels even after the Wortmannin treatment. Perifosine did not significantly enhance the apoptosis induction by GT-7 in the three PDAC cells (Figure 4A). This is also presumably due to the remaining AKT phosphorylation levels by GT-7 treatment. Thus, GT-7-induced AKT phosphorylation appears to have an antagonistic effect on the inhibitory action of both Wortmannin and Perifosine in PANC-1 cells.

Bondar et al. reported that AsPC-1 cells (one of the PDAC cell lines) are resistant to Wortmannin-induced apoptosis and that Wortmannin only marginally down-regulated phospho-AKT levels in AsPC-1. In contrast, in the cells that underwent apoptosis by Wortmannin, complete down-regulation of phosphorylation was observed [17]. This situation is also found in our data that Wortmannin and Perifosine did not significantly reduce the AKT phosphorylation levels (see Figure 4B, quantification of pAKT/tAKT) when co-treated with GT-7. This residual activity may inhibit apoptosis induction by these PI3K/AKT inhibitors.

Why AKT phosphorylation in PANC-1 cells is resistant to Wortmannin treatment remains unknown. However, a recent MS proteomics combined with enrichment GO categories identified the protein phosphatase 2A (PP2A) complex with a key role in the viability of PANC-1 cells [18]. The expression of PP2A complex is markedly lower in PANC-1 in comparison with MIA-Pa-Ca-2, and PP2A activation with DT-061 was able to suppress the viability of PANC-1. Consistently, siRNA knockdown of the PP2A subunit allowed higher viability of PANC-1 cells. Thus, the residual AKT phosphorylation and sustained viability of PANC-1 cells upon GT-7 treatment may be associated with a lower expression profile of the PP2A complex.

To further evaluate the effect of Wortmannin on GT-7-mediated cell death, FCM analysis was performed on the three PDAC cells using 10 μM GT-7 in combination with Wortmannin. As shown in Figure 4C, Wortmannin treatment significantly increased cell death population in MIA-Pa-Ca-2 (from 40.1% to 54.0%) and T3M4 (from 50% to 70.3%) but not in PANC-1 (21.2% to 20.7%), which is consistent with the data obtained with the quantification of cleaved Caspase-3. Phase-contrast images of the PDAC cells treated with GT-7 reflect the cell death status using immunoblot against cleaved Caspase-3 (Appendix A).

To further investigate the impact of the AKT signaling activation on GT-7-mediated apoptosis, we examined the effect of PTEN silencing on GT-7-induced apoptosis in T3M4 cells. PTEN is a major negative regulator of the PI3K/AKT signaling, and its knockdown will lead to AKT activation. As shown in Figure 4D, the PTEN siRNAs effectively suppressed the targeted gene expression at the protein levels in T3M4 cells in comparison with the control siRNA. Consistently, AKT phosphorylation levels were increased by PTEN silencing, and GT-7 further increased AKT phosphorylation levels in the PTEN silenced conditions (Figure 4D,E). Notably, PTEN silencing significantly attenuated the GT-7-mediated apoptosis induction as evaluated by the relative quantification of cleaved Caspase-3 (Figure 4D,E). Thus, stimulating AKT phosphorylation levels by PTEN silencing counteracted the effect of GT-7 to induce apoptosis in T3M4 cells, indicating that AKT phosphorylation levels can affect the susceptibility of T3M4 to GT-7-induced cell death.

Altogether, the AKT phosphorylation levels in each PDAC strain are likely to play key roles in dictating the susceptibility to GT-7-mediated apoptosis.

## 4. Discussion

Pancreatic ductal adenocarcinoma (PDAC) is the most common type of pancreatic malignancy, accounting for approximately 90% of all pancreatic cancers. KRAS mutations, present in about 95% of pancreatic cancers, lead to the robust activation of the ERK MAPK and AKT signaling. Thus, targeting the MAPK and/or AKT signaling driven by oncogenic KRAS appears to be a valuable approach for PDAC therapy. However, clinically available inhibitors for the ERK and AKT signaling are still limited, with no superior effect in pancreatic cancer patients in comparison with conventional cytotoxic chemotherapy. Given the poor clinical outcome of patients with PDAC, together with its resistance to chemoradiotherapy, novel strategies are needed to introduce new treatment opportunities. In this study, we have shown that GT-7, an anti-cancer compound with ERK modulating property, can provide a novel approach to tackle this deadly malignancy.

### 4.1. The Potential and Significance of GT-7 as a Compound to Induce ERK-Dependent Apoptosis in PDAC

GT-7 decreased cell viability and induced apoptosis in the three PDAC cell lines with different KRAS mutations. The effect of GT-7 on cell viability and apoptosis induction was much superior to Honokiol, a natural agent known to be effective in cancer therapy, including PDAC. The uniqueness of GT-7 as an inducer of apoptosis relies on its property to stimulate ERK phosphorylation. Notably, in T3M4 cells harboring the KRAS Q61H mutation, ERK activation stimulated by GT-7 is required for apoptosis induction, as evidenced by the significant cancellation of apoptosis induction by the MEK inhibitor U0126. A growing number of the literature showed that more than 50 compounds have been reported to induce ERK-dependent cell death in various tumor cell types [8,9]. However, thus far, no reports have been made on an anti-cancer compound capable of inducing ERK-dependent apoptosis. What types of tumor or oncogenic context is more susceptible to ERK-dependent cell death? GT-7 and its original compound ACA-28 were shown to induce ERK-dependent apoptosis in melanoma cell lines, harboring oncogenic BRAF or NRAS mutation, but not in the normal melanocyte (NHEM) [6]. In addition, ACA-28 was shown to induce ERK-dependent apoptosis in NIH/3T3 cells overexpressing active HER2, but not in NIH/3T3 cells [6]. Thus, ACA-28 and/or GT-7 were shown to induce apoptosis in cancer cell lines with different oncogenic mutations, ranging from HER2, BRAF, NRAS, and KRAS. Since one of the hallmarks of ERK-induced cell death is “sustained ERK activation”, it seems plausible that cells harboring high-ERK with oncogenic alterations in upstream activators of ERK signaling are more susceptible to apoptosis induction by GT-7. Consistently, several papers have provided the hypothesis on a cellular threshold for active ERK1/2 levels in determining the cell fate for growth arrest versus death responses mediated by ERK1/2 signaling. Thus, sustained ERK1/2 activation above a certain threshold may activate specific pro-apoptotic targets and induce apoptosis [8,9].

Although precise molecular mechanisms of how phospho-ERK induces cell death remain elusive, a large number of agents have been reported to induce ERK activation-dependent apoptosis through DNA-damaging stimuli-mediated p53 activation [8,9]. Several papers reported that apoptosis induction by various DNA-damaging agents correlates with p53 up-regulation, suggesting that ERK activation stimulates p53 transcriptional activity, which will lead to the up-regulation of pro-apoptotic genes, including Bcl-2, Bax, and Noxa [8,9]. Alternatively, many compounds capable of inducing ERK-mediated apoptosis share the property to stimulate reactive oxygen species (ROS) [8,9]. ROS can stimulate ERK activity either by activation of EGFRs or by inhibition of MAPK phosphatases (MKPs or DUSPs). ROS can attack the cysteine residues of a certain group of target proteins, thereby oxidizing the reactive thiol groups to form a disulfide bond. Oxidation of catalytic cysteine within the catalytic site of the DUSPs abolishes phosphatase activities of DUSPs as well as several tyrosine phosphatases that inactivate upstream regulators of ERK signaling [8,9]. Indeed, ROS have been shown to block phosphatase activities of ERK-directed phosphatases, DUSP5 and DUSP6, by oxidation of their catalytic cysteine residues [8,9]. Moreover, ROS accumulation stimulates ERK activation, which triggers proteasomal degradation of DUSPs. Interestingly, a recent paper by Song et al. showed that knockdown of Synaptophysin-like 1(SYLP1), a neuroendocrine-like protein, induced cell death by stimulating sustained ERK activation. SYLP1 knockdown was shown to induce ROS, which led to ERK activation and cell death [19]. Thus, a similar ROS-mediated ERK-dependent cell death mechanism might be associated with GT-7 action.

It should be noted that the GT-7-mediated increase in pERK was blocked by the MEK inhibitor U0126 in all three PDAC, but apoptosis was inhibited by U0126 only in T3M4 cells, not MIA-Pa-Ca-2 or PANC-1 cells (Figure 2). What is the possible reason for the differential effect of pERK on apoptosis induction in these PDAC cell lines? Several papers reported that not only the kinetics of ERK phosphorylation (transient versus sustained) but also the spatial distribution of phosphorylated ERK and its substrates play a critical role in determining the fate of the cells for anti- versus pro-apoptosis. Importantly, scaffold proteins, including DAPK and PEA-15, have been reported to regulate the spatial distribution of phosphorylated ERK by anchoring the phosphorylated ERK protein in the cytosol [20,21]. For example, activated ERK1/2 is sequestered in the cytoplasm via interaction with PEA-15 and DAPK. Inhibition of ERK1/2 nuclear translocation impairs ERK1/2-mediated proliferation and augments the pro-apoptotic signals of DAPK by phosphorylating the cytoplasmic DAPK. In addition, DUSPs also play key roles in determining the distribution of phosphorylated ERK1/2. DUSP6 serves as an anchor for inactive ERK in the cytosol [22]. Interestingly, ROS produced by various compounds capable of inducing ERK-dependent apoptosis inactivates the cytosolic ERK phosphatase DUSP6, resulting in cytoplasmic sequestration of active ERK [23]. Thus, it would be intriguing to speculate that these docking phosphatases (DUSPs) and/or scaffold proteins may be differentially expressed in these PDACs and that T3M4 possesses the most favorable conditions in sustaining ERK phosphorylation distribution to induce apoptosis.

How does GT-7 increase pERK? ACA has been reported as an inhibitor for nuclear export of Rev by binding to the Cysteine-529 residue of CRM1, the receptor for NES, thereby inhibiting nuclear export of Rev [24]. Tamura et al. reported the formation of the quinone methide intermediate of ACA to be essential for exerting the inhibitory activity of ACA for nuclear export of Rev [25]. Furthermore, the treatment of ACA with N-Acetyl-cysteine (NAC) was found to provide the two adducts. As ACA-28 and GT-7 bear a similar structure (two carbonate esters) to ACA regarding the binding to NAC, it is highly probable that ACA-28 and GT-7 via the formation of the quinone methide intermediate, thereby serving as oxidants to attack the Cysteine residues of the target proteins. Consistently, ACA, the original compound, has been reported to induce ROS [26]. In ACA- and sodium butyrate-treated cells, intracellular ROS levels and NADPH oxidase activities were increased in HepG2 human hepatocellular carcinoma cells. The decrease in cell number after combined treatment of ACA and sodium butyrate was diminished when cells were pretreated with catalase, a strong antioxidant enzyme that breaks down ROS. The authors also showed that ACA, alone or in combination with sodium butyrate, increased pERK. Similar observations of the ACA-mediated induction of ROS and the inhibition of apoptosis induction by a thiol antioxidant NAC have been reported in other cancer cell lines such as NB4 promyelocytic leukemia cells [27]. We, therefore, assume that GT-7 and ACA-28, similar to the original compound ACA, can increase pERK via stimulation of ROS signaling. As mentioned previously, targets of ROS include various phosphatases that inactivate upstream kinases of ERK signaling as well as phosphatases against ERK. Thus, both scenarios remain possible regarding the action of GT-7 in that this agent can increase MEK activity or reduce phosphatase activity against pERK depending on the context of the genome or gene profiling expression in each cancer cell line.

### 4.2. GT-7-Mediated Apoptosis Can Be Enhanced by AKT Signaling Inhibition

Importantly, our study identified AKT signaling as a key to providing resistance to GT-7-mediated apoptosis. The three PDAC cell lines examined in this study displayed different responses to GT-7 in terms of apoptosis susceptibility. We showed that sustained AKT phosphorylation correlates with resistance to GT-7-induced apoptosis (Figure 3). The susceptibility of induction of apoptosis is stronger in T3M4 than in MIA-Pa-Ca-2 and weakest in PANC-1, as evidenced by the quantification of relative cleaved Caspase-3/full-length Caspase-3 and FCM analysis. As mentioned above, sustained AKT phosphorylation correlates with resistance to GT-7-induced apoptosis. PANC-1 displayed the most sustained AKT phosphorylation, followed by MIA-Pa-Ca-2. T3M4 cells displayed marked down-regulation of AKT phosphorylation levels below the basal levels after a transient increase in the AKT phosphorylation 2 h after the GT-7 treatment. Consistently, in our previous study, GT-7 and the original compound ACA-28 induced ERK-dependent apoptosis in SK-MEL-28, an ERK-active melanoma cell line, wherein AKT phosphorylation levels are low. These observations, together with the effect of Wortmannin and PTEN silencing, suggest that AKT signaling activation plays a key role in determining susceptibility to GT-7.

What is the mechanism of action for GT-7 in increasing both pERK and pAKT? We assume that ROS could be involved in the GT-7 action based on the findings that a large number of compounds capable of inducing ERK-dependent cell death elicits ROS signaling. As described in the previous manuscript, ROS-mediated DUSP inactivation/degradation is a part of the mechanism to stimulate ERK signaling. Furthermore, ROS facilitated cell death through the activation of AKT. Chetram et al. observed that ROS enhanced the expression of phosphorylated AKT (pAKT) in human prostate cancer cells [28]. Notably, this ROS-mediated AKT activation is through PTEN, a negative regulator of AKT signaling, because PTEN was rendered catalytically inactive through oxidation by ROS. Importantly, the two vanadium compounds, bis(acetylacetonato)-oxidovanadium(IV) and sodium metavanadate, inhibit cell proliferation via ROS-induced activation of both PI3K/AKT and MAPK/ERK signaling pathways in human pancreatic cancer AsPC-1 cells [29]. The effects of the two compounds could be counteracted with the antioxidant NAC. Thus, GT-7 could induce ROS, thereby increasing both pERK and pAKT.

A plausible underlying mechanism explaining the different AKT signaling activation status in the three PDAC cell lines may rely on the distinct KRAS mutations in the three PDAC. T3M4 cells harbor KRAS Q61H, whereas PANC-1 cells harbor KRAS G12D, and MIA-Pa-Ca-2 cells harbor KRAS G12C. Wild-type KRAS transmits signals through ERK and AKT signaling pathways via interaction with RAF and PI3K, respectively. Importantly, KRAS Q61H displays preferential binding to RAF compared with PI3K, leading to enhanced MAPK signaling [30]. Moreover, the KRAS G12D mutant found in PANC-1 preferably activates AKT signaling as compared with ERK signaling [31,32]. The KRAS G12C mutant found in MIA-Pa-Ca-2 strongly binds with RAF1, and AKT signaling activation was low [33]. These previous findings in the literature and our data on the responses of AKT signaling activation upon GT-7 seem to consistently explain the basis of cells’ resistance to GT-7. Regarding other possible oncogenic mutations in PDAC, the three PDAC cell lines share common genomic sequences in that they harbor the wild-type PTEN and CDKN1A and the mutant TP53. However, special caution is needed regarding the effect of these KRAS mutations on ERK and AKT signaling as the effect may be variable depending on the cell lines and the number of alleles as well as other mutations in the genome in each cell line.

Regarding the susceptibility of inhibition of cell viability is stronger in MIA-Pa-Ca-2, intermediate in T3M4, and weaker in PANC-1. Thus, PANC-1 displayed the strongest resistance both in terms of WST-8 assay and cell death evaluation. The reversal of susceptibility between MIA-Pa-Ca-2 and T3M4 may indicate that GT-7 may affect growth inhibitory mechanisms other than apoptosis induction. The possible growth inhibitory mechanisms include cell cycle arrest or autophagy induction. Thus, the analysis of the effect of GT-7 (especially at lower concentrations) on the cell cycle and/or autophagy in MIA-Pa-Ca-2 in comparison with other PDAC cell lines may reveal a novel mode of action of GT-7.

An additional factor that should be taken into consideration, relevant to cells’ responses to GT-7, is the epithelial-mesenchymal phenotype because epithelial-mesenchymal transition (EMT) is closely associated with enhanced migration, invasion, and resistance to anti-cancer chemotherapy. Epithelial cells expressing epithelial markers such as E-cadherin maintain apical-based polarity and sheet-like growth architecture with tight junctions. Epithelial cells undergo a phenotypical shift and acquire mesenchymal features, including spindle-like fusiform, motile cells, which express mesenchymal markers such as vimentin and N-cadherin. Minami et al. reported important findings regarding the morphofunctional analysis of various PDAC cell lines using 2- and 3-dimensional cultures. Because PDAC cells form tumors with 3-dimensional structures, these provide valuable information on cells’ morphological features and their impact on functional and pathological aspects [34]. Based on this paper, MIA-Pa-Ca-2 and PANC-1 displayed irregular-shaped spheres with small oval cells, and the spheres had a grape-like morphology without flat-lining cells, whereas T3M4 cells displayed small oval and flat-lining cells on the surface of the spheres. These morphological features indicate that PANC-1 and MIA-Pa-Ca-2 cells displayed mesenchymal features, whereas T3M4 cells displayed epithelial features. Moreover, regarding the EMT phenotype, T3M4 had high E-cadherin and low vimentin mRNA levels, while PANC-1 and MIA-Pa-Ca-2 had low E-cadherin and high vimentin mRNA levels. Thus, the EMT-positive cell lines displayed relative resistance to the GT-7-induced apoptosis, whereas the EMT-negative T3M4 displayed a stronger sensitivity to apoptosis induction by GT-7. Furthermore, the EMT score evaluated by Young Kwang Chae’s method [18], based on the statistical analysis of the gene expression profiling, showed that T3M4 showed a strong epithelial phenotype, while MIA-Pa-Ca-2 and PANC-1 displayed mesenchymal or quasi-mesenchymal phenotype, consistent with the 3D analysis. An increasing amount of evidence has indicated that EMT is closely associated with chemotherapy resistance through the involvement of EMT-associated transcription factors in various human cancers. Therefore, cells with an EMT phenotype may be more likely to be resistant to apoptosis induction by GT-7.

Furthermore, we investigated the genome-wide gene expression profile utilizing the Cancer Cell Line Encyclopedia (CCLE: https://sites.broadinstitute.org/ccle/, accessed on 29 August 2021) and qRT-PCR, aiming to reveal the basis for responses of the three PDAC cell lines to GT-7. We searched for factors in the apoptosis pathway responsible for the responses to the sensitivity to the agent from the CCLE database. As a result, we identified several apoptosis-related genes that are highly expressed in MIA-Pa-Ca-2 and T3M4 but poorly expressed in PANC-1 (Appendix A). These include EREG, encoding epiregulin, and ANXA1, encoding Annexin 1 (Appendix A). Epiregulin is important for the EGFR-mediated growth signaling, and Annexins 1 attracts attention for its prognostic significance in PDAC [35]. Epiregulin belongs to the epidermal growth factor (EGF) family of ligands that play important roles in the pathology of PDAC through the activation of the MEK/ERK signaling pathway [36]. Annexin A1 (ANXA1) has been reported to promote apoptosis in cancerous cells, including PDAC [37]. Notably, reduced expression of Annexin A1 has been reported to promote gemcitabine and 5-fluorouracil drug resistance of human pancreatic cancer [38]. These data are consistent with our data that MIA-Pa-Ca-2 and T3M4 are relatively sensitive to GT-7-mediated apoptosis, whereas PANC-1 showed strong resistance against the agent.

As mentioned earlier, pAKT levels affected apoptotic responses to GT-7 in these PDAC cell lines. However, considering the threshold theory proposed by Park’s lab, it may be that in order to achieve ERK-dependent apoptosis, the higher cellular ERK phosphorylation levels would be more favorable. In this regard, differential gene expression patterns of EREG and ANXA1, which are highly expressed in MIA-Pa-Ca-2 and T3M4 but poorly expressed in PANC-1, might contribute to the more sustained ERK signaling activation preferable to achieve GT-7-mediated apoptosis. Whether epiregulin and annexin A1 may serve as factors that provide differential responses of the three PDAC cell lines to this agent needs future validation.

### 4.3. Perspectives

PI3K/AKT and ERK MAPK pathways regulate each other’s activity through feedback mechanisms. For example, MEK inhibition by U0126 potentiates EGF- and FGF-induced AKT phosphorylation [39,40]. In addition, dual inhibition of PI3K/mTOR induces overactivation of the MEK/ERK pathway in human pancreatic cancer [41]. Therefore, the inefficiency of single-agent treatments of these promising targeted therapies has been attributed partly due to the rapid up-regulation of compensatory alternative pathways and feedback loops within tumor cells, which gives a rationale for the concurrent targeting of both pathways [42]. Consistently, several reports have been made on simultaneous treatment with two pathway inhibitors. These include BEZ235 (dual PI3K/mTOR kinase inhibitor) and the MEK inhibitor PD0325901, and combined targeting of RAF/MEK/ERK signaling and autophagy survival responses [2] in PDAC. In this regard, our study, by demonstrating the effective apoptosis induction by the combination of GT-7 (ERK stimulation) and Wortmannin (PI3K inhibition), proposes a potential novel approach for concurrent targeting of ERK and PI3K/AKT signaling pathways. In addition, these findings may provide novel information on how to induce apoptosis depending on the different KRAS mutation alleles in PDAC.

## Figures and Tables

**Figure 1 cells-11-00702-f001:**
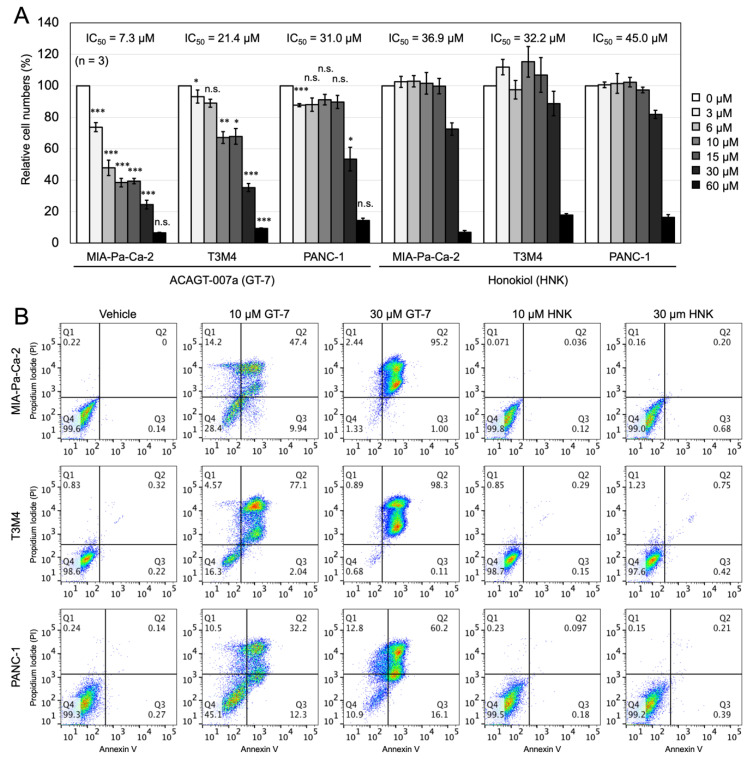
Effects of ACAGT-007a (GT-7) on cell viability and apoptosis of the PDAC cells. (**A**) Comparison of the cell viability upon GT-7 or Honokiol (HNK) in the PDAC cells. The PDAC cells were treated by GT-7 or Honokiol at serial concentrations (0–60 μM) for 48 h, then the relative living cell numbers (%) were measured by WST-8 assay. The data were averaged from three independent experiments (*n* = 3). Columns, means; bars, SEM. * *p* < 0.05, ** *p* < 0.01, *** *p* < 0.005, n.s., not significant; significantly different between GT-7 and Honokiol in each concentration using a Student’s *t*-test. (**B**) Effect of GT-7 and Honokiol (HNK) on apoptosis of PDAC cells. The PDAC cells were treated by GT-7 or Honokiol at the concentration indicated for 24 h, and then apoptotic cells were analyzed by FCM.

**Figure 2 cells-11-00702-f002:**
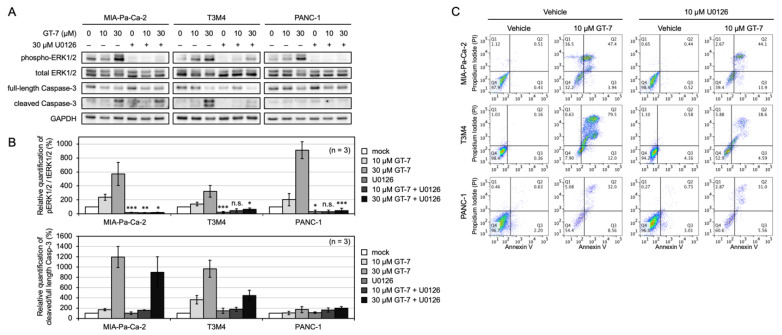
GT-7-induced apoptosis is dependent on ERK activation in T3M4 but not in MIA-Pa-Ca-2 and PANC-1 cells. (**A**) MEK inhibitor U0126 attenuates GT-7-induced apoptosis in T3M4 cells but not MIA-Pa-Ca-2 and PANC-1 cells. The PDAC cells were treated by GT-7 and/or U0126 for 2 h. The indicated proteins were detected by Western blot analysis. (**B**) Relative quantification of ERK phosphorylation levels and cleaved/full-length Caspase-3 in PDAC cells upon GT-7 and/or U0126 treatment as shown in (**A**). Phosphorylation levels (phosphorylated protein intensity/total protein intensity) after the treatment with DMSO (0 μM GT-7 without 30 μM U0126) in each PDAC cell were set as 100%. The data were averaged from three independent experiments (*n* = 3). Columns, means; bars, SEM. * *p* < 0.05, ** *p* < 0.01, *** *p* < 0.005, n.s., not significant. Comparisons between absence and presence of 30 μM U0126 were made by a Student’s *t*-test. (**C**) Apoptosis induced by GT-7 was suppressed by the MEK inhibitor U0126 in T3M4 cells but not in MIA-Pa-Ca-2 and PANC-1 cells. MIA-Pa-Ca-2, T3M4, and PANC-1 cells were treated with DMSO (vehicle), 10 μM GT-7, 10 μM U0126, or the combination of 10 μM GT-7 and 10 μM U0126 for 24 h. Chemical treated cells were stained by Annexin V-FITC and PI and analyzed by FCM.

**Figure 3 cells-11-00702-f003:**
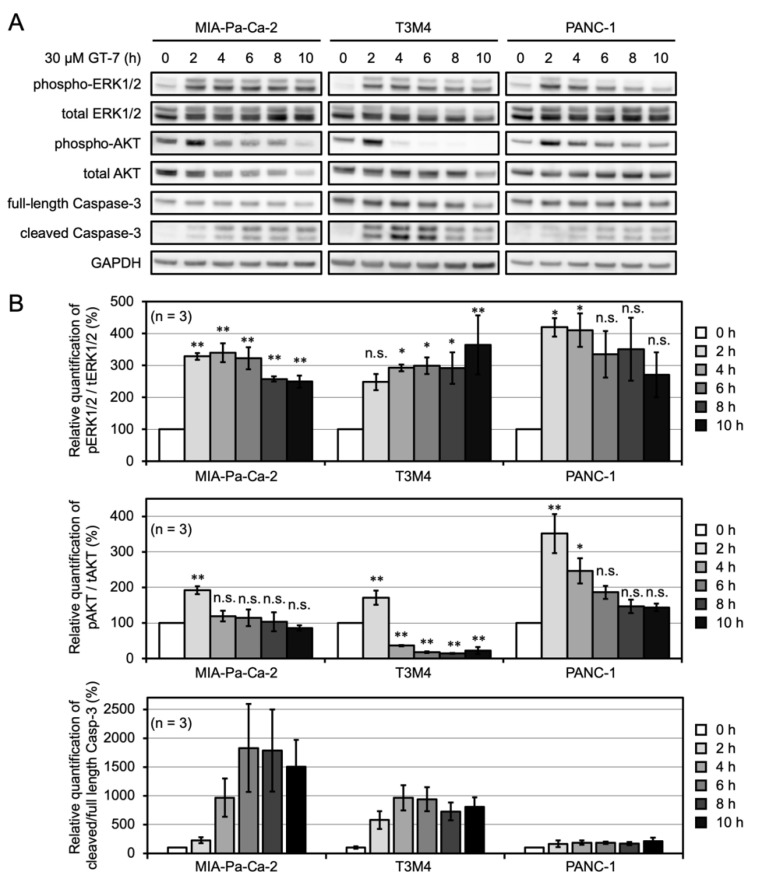
Apoptosis induction and the activation of both ERK and AKT signaling by ACAGT-007a (GT-7) in the PDAC cells. (**A**) The PDAC cells were treated with 30 μM GT-7 for the time indicated (0–10 h). The indicated proteins were detected by Western blot analysis. (**B**) Relative quantification of ERK and AKT phosphorylation levels and cleaved/full-length Caspase-3 in the PDAC cells upon GT-7 treatment as shown in (**A**). Phosphorylation levels (phosphorylated protein intensity/total protein intensity) after the treatment with DMSO (0 h) in each PDAC cell were set as 100%. The data were averaged from three independent experiments (*n* = 3). Columns, means; bars, SEM. * *p* < 0.05, ** *p* < 0.01, n.s., not significant; significantly different from DMSO (0 h) in each cell line using one-way ANOVA, followed by a post hoc test using Dunnett’s multiple comparisons.

**Figure 4 cells-11-00702-f004:**
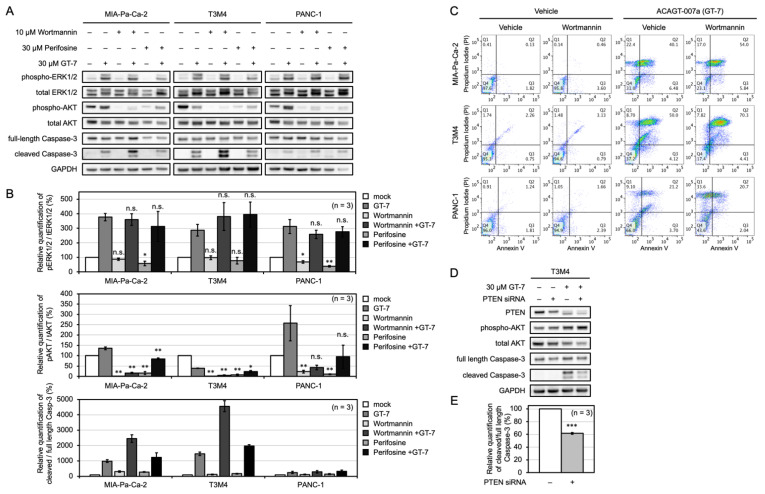
AKT inhibition promotes ACAGT-007a (GT-7)-induced apoptosis in PDAC cells. (**A**) The effect of PI3K/AKT inhibitors (Wortmannin and Perifosine) on the induction of GT-7-induced apoptosis in the PDAC cells. The PDAC cells were pretreated by 30 μM Wortmannin or Perifosine for 1 h then treated by 30 μM GT-7 for 3 h. The indicated proteins were detected by Western blot analysis. (**B**) Relative quantification of ERK and AKT phosphorylation levels and cleaved/full-length Caspase-3 in the PDAC cells upon PI3K/AKT inhibitors pretreatment and GT-7 treatment as shown in (**A**). Phosphorylation levels (phosphorylated protein intensity/total protein intensity) after the treatment with DMSO (without compounds) in each cell were set as 100%. The data were averaged from three independent experiments (*n* = 3). Columns, means; bars, SEM. * *p* < 0.05, ** *p* < 0.01, *** *p* < 0.005, n.s., not significant. Significant differences from the DMSO treatment (without PI3K/AKT inhibitors) in each absence or presence of GT-7 using one-way ANOVA, followed by a post hoc test using Dunnett’s multiple comparisons (upper and middle panels). Comparisons between absence and presence of GT-7 were made by a Student’s *t*-test (lower panel). (**C**) The inhibition of AKT phosphorylation by Wortmannin enhances GT-7 induced apoptosis in MIA-Pa-Ca-2 and T3M4 cells but not in PANC-1 cells. PDAC cells were treated with DMSO (vehicle), 10 μM GT-7, 10 μM Wortmannin, or the combination of 10 μM GT-7 and 10 μM Wortmannin for 6 h. Chemical treated cells were stained by Annexin V-FITC and PI and analyzed by FCM. (**D**) The cleaved Caspase-3 was reduced by PTEN knockdown in T3M4 cells. T3M4 cells were pretreated by the negative control siRNA or PTEN siRNA for 48 h then treated by 30 μM GT-7 for 2 h. The indicated proteins were detected by Western blot analysis. (**E**) Relative quantification of cleaved Caspase-3 was averaged from three independent experiments (*n* = 3). Columns, means; bars, SEM. *** *p* < 0.005, significantly different between the treatment with negative control siRNA (PTEN siRNA−) and the treatment with PTEN siRNA (PTEN siRNA +) then GT-7 treatment for 2 h using a Student’s *t*-test.

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
