# Peer review of "ACAGT-007a, an ERK MAPK Signaling Modulator, in Combination with AKT Signaling Inhibition Induces Apoptosis in KRAS Mutant Pancreatic Cancer T3M4 and MIA-Pa-Ca-2 Cells"

_cells, 2022, doi:10.3390/cells11040702_

Round 1

Reviewer 1 Report

The authors answered all my questions, performed additional experients and provided the requested quantifications.

Just two minor remarks:

  • the color code for the bar graphs in Figure 4B seems wrong
  • I'm sceptical about reference 23 in the discussion. It is not common sense that KRAS G12D would not signal through ERK (compare e.g. Ihle et al., Natl Cancer Inst, 2012)

Author Response

We are grateful to Reviewer 1 for providing useful suggestions and insightful comments that helped us to considerably improve our manuscript.  As indicated in the following responses, we have incorporated these comments and suggestions in our revised manuscript.

              The reviewer made the following observations: The authors answered all my questions, performed additional experients and provided the requested quantifications.

Reviewer 1 Comments for the Author:

  1. The reviewer mentioned, “the color code for the bar graphs in Figure 4B seems wrong”.

Re: Accordingly, we amended the error in the new manuscript in Figure 4B.

  1. The reviewer mentioned, “I'm sceptical about reference 23 in the discussion. It is not common sense that KRAS G12D would not signal through ERK (compare e.g. Ihle et al., Natl Cancer Inst, 2012)

Re: We thank Reviewer 1 for the pertinent comment and suggestion for the reference on the KRAS G12D mutation analyzed using the data from a clinical trial for patients with non-small cell lung cancer (NSCLC) (Ihle et al., 2012), which describes that the KRAS G12D mutation preferably activates AKT signaling, whereas wild-type KRAS activation results in signaling through MEK, AKT, and RalA/B (Figure 6: Summary). Thus, the effect on KRAS G12D on ERK signaling seems controversial, although a recent study by Lentch et al., showed that knocking-out of the KRAS G12D protein in Panc-1 cell resulted in comparatively stable levels of pERK, whereas pAKT levels were reduced in some KRAS clones (Lentch et al., 2019). We, therefore, revised the description that “the KRAS G12D mutation preferably activates AKT signaling as compared with ERK signaling” and incorporate these descriptions and references with special caution in the revised manuscript, mentioning that the effect of the KRAS G12D mutation on ERK and AKT signaling may be variable dependent on the cell lines and the number of alleles as well as other mutations in the genome in each cell line.

Reviewer 2 Report

  1. This is a manuscript that has been extensively revised in both text and Figures, some that were deleted. A major make-over that has changed the presentation in response to criticisms, but has not necessarily remedied the noted weaknesses. This team of investigators from Kindai University in Osaka and the topic of cancer cell biology are appropriate for the special issue.
  2. This report is focused on GT-7, a derivative of a compound previously found in a chemical library for the ability to induce apoptosis in melanoma cell lines. Here GT-7 is tested in three pancreatic cell lines (PANC) that harbor different KRAS mutations. All three cell lines showed reduced viability to GT-7 with different EC50 in the low micromolar range. This reduced viability was attributed to an increase in apoptosis, shown by flow cytometry and increased annexin V staining. In both viability and apoptosis assays T3M4 cells with KRAS Q61H were most sensitive to GT-7, compared to MIA-Pa-Ca-2 cells with G12C, while PANC-1 with G12D were least sensitive (Fig 1). These data are clearly presented and the experiments appear to be carefully done.
  3. GT-7 increased the level of phosphorylation of ERK in all 3 PANC, determined by immunoblotting. This increased pERK was blocked by the MEK inhibitor UO126 in all three PANC, but apoptosis (assayed by Caspase-3 cleavage) was inhibited by UO126 only in T3M4 cells, not MIA-Pa-Ca-2 or PANC-1 cells (Fig. 2). This is a curious observation that lacks a satisfying explanation as to how pERK is coupled to apoptosis differently in these PANC. Further, what substrate(s) of sustained pERK might be responsible for triggering apoptosis, beyond the speculation in lines 450-457?
  4. Phosphorylation of S473 in AKT was increased in all 3 PANC in response to GT-7, but each cell line showed different duration of response, with a loss of signal in T3M4 cells. (Fig 3). The sustained elevation of pAKT after GT-7 is correlated to resistance to apoptosis. A reasonable correlation, but there are no insights into mechanism of action for GT-7 in increasing both pERK and pAKT. Wortmannin to inhibit PI3K and reduce pAKT enhanced apoptosis in T3M4 and MIA-Pa-Ca-2 cells, but not PANC-1 cells (Fig 4). On the other hand Perifosine, an AKT inhibitor did not enhance GT-7 induced apoptosis in any of the PANCs. Silencing PTEN in T3M4 cells elevated pAKT and attenuated GT-7 induced apoptosis. Overall the conclusion was that pAKT levels affected apoptotic responses to GT-7 in PANC.
  5. What remains unclear is the actual target of this drug, and therefore its mechanism of action in increasing pERK. How does GT-7 increase pERK is a fundamental and critical question. Does GT-7 increase MEK activity, or reduce phosphatase activity against pERK? These critical questions are not addressed, and maybe beyond the scope of these studies. If so, it leaves this as a rather descriptive study, with observations about the effects of different compounds on pERK, pAKT and apoptosis, but not providing mechanistic insights that would greatly increase impact and significance. We know no more about how GT-7 works after this study than before.
  6. There is some discussion about the difference in signaling by the various KRAS mutants present in the PANC cell lines (lines 491-505). But, how does this account for the greater sensitivity to GT-7 in the Q61H KRAS cells and relative resistance of the G12D KRAS cells? The association with gene expression described in lines 542-560 is highly speculative and untested by experiments.

Author Response

We are grateful to Reviewer 2 for providing useful suggestions and insightful comments that helped us to considerably improve our manuscript.  As indicated in the following responses, we have incorporated these comments and suggestions in our revised manuscript.

Reviewer 2 Comments for the Author:

  1. The reviewer mentioned, “This is a manuscript that has been extensively revised in both text and Figures, some that were deleted. A major make-over that has changed the presentation in response to criticisms, but has not necessarily remedied the noted weaknesses. This team of investigators from Kindai University in Osaka and the topic of cancer cell biology are appropriate for the special issue.

Re: We thank Reviewer 2 for positive responses to our revised manuscript.  We further revised our manuscript by incorporating your comments and concerns, which we believe, considerably improved our manuscript.

  1. The reviewer mentioned, “This report is focused on GT-7, a derivative of a compound previously found in a chemical library for the ability to induce apoptosis in melanoma cell lines. Here GT-7 is tested in three pancreatic cell lines (PANC) that harbor different KRAS mutations. All three cell lines showed reduced viability to GT-7 with different EC50 in the low micromolar range. This reduced viability was attributed to an increase in apoptosis, shown by flow cytometry and increased annexin V staining. In both viability and apoptosis assays T3M4 cells with KRAS Q61H were most sensitive to GT-7, compared to MIA-Pa-Ca-2 cells with G12C, while PANC-1 with G12D were least sensitive (Fig 1). These data are clearly presented and the experiments appear to be carefully done.

  1. The reviewer mentioned, “GT-7 increased the level of phosphorylation of ERK in all 3 PANC, determined by immunoblotting. This increased pERK was blocked by the MEK inhibitor UO126 in all three PANC, but apoptosis (assayed by Caspase-3 cleavage) was inhibited by UO126 only in T3M4 cells, not MIA-Pa-Ca-2 or PANC-1 cells (Fig. 2). This is a curious observation that lacks a satisfying explanation as to how pERK is coupled to apoptosis differently in these PANC. Further, what substrate(s) of sustained pERK might be responsible for triggering apoptosis, beyond the speculation in lines 450-457?

Re: We thank Reviewer 2 for these pertinent comments. First, as the reviewer suggested, the differential effect of pERK on apoptosis induction in these PDAC cell lines is an interesting issue, which deserves mentioning with a plausible explanation. In addition, substrate(s) of sustained pERK that might be responsible for triggering apoptosis would be an important, as-yet clarified issue in the field of ERK-dependent apoptosis. We, therefore, added the following sentence in the Discussion section.

What is the possible reason for the differential effect of pERK on apoptosis induction in these PDAC cell lines? Several papers reported that not only the kinetics of ERK phosphorylation (transient versus sustained) but also the spatial distribution of phosphorylated ERK and its substrates plays a critical role in determining the fate of the cells for anti- versus pro-apoptosis. Importantly, scaffold proteins, including DAPK and PEA-15 have been reported to regulate the spatial distribution of phosphorylated ERK by anchoring the phosphorylated ERK protein in the cytosol (Chen et al., 2005; Formstecher et al., 2001). For example, activated ERK1/2 is sequestered in the cytoplasm via interaction with PEA-15 and DAPK. Inhibition of ERK1/2 nuclear translocation impairs ERK1/2-mediated proliferation and augments the pro-apoptotic signals of DAPK by phosphorylating the cytoplasmic DAPK. In addition, DUSPs also play key roles in determining the distribution of phosphorylated ERK1/2. DUSP6 serves as an anchor for inactive ERK in the cytosol (Bermudez et al., 2010). Interestingly, ROS produced by various compounds capable of inducing ERK-dependent apoptosis, inactivate the cytosolic ERK phosphatase DUSP6, resulting in cytoplasmic sequestration of active ERK (Kim et al., 2003).

Thus, it would be intriguing to speculate that these docking phosphatases (DUSPs) and/or scaffold proteins may be differentially expressed in these PDACs and that T3M4 possesses the most favorable conditions in sustaining ERK phosphorylation distribution to induce apoptosis.

  1. The reviewer mentioned, “Phosphorylation of S473 in AKT was increased in all 3 PANC in response to GT-7, but each cell line showed different duration of response, with a loss of signal in T3M4 cells. (Fig 3). The sustained elevation of pAKT after GT-7 is correlated to resistance to apoptosis. A reasonable correlation, but there are no insights intomechanism of action for GT-7 in increasing both pERK and pAKT. Wortmannin to inhibit PI3K and reduce pAKT enhanced apoptosis in T3M4 and MIA-Pa-Ca-2 cells, but not PANC-1 cells (Fig 4). On the other hand Perifosine, an AKT inhibitor did not enhance GT-7 induced apoptosis in any of the PANCs. Silencing PTEN in T3M4 cells elevated pAKT and attenuated GT-7 induced apoptosis. Overall the conclusion was that pAKT levels affected apoptotic responses to GT-7 in PANC.

Re: We thank Reviewer 2 for these pertinent comments. Regarding the mechanism of action for GT-7 in increasing both pERK and pAKT, we assume that ROS could be involved in the GT-7 action, based on the findings that a large number of compounds capable of inducing ERK-dependent cell death elicits ROS signaling. As described in the previous manuscript, ROS-mediated DUSP inactivation/degradation is a part of the mechanism to stimulate ERK signaling (in the Discussion section). Furthermore, ROS facilitated cell death through the activation of AKT. Chetram et al. observed that ROS enhanced the expression of phosphorylated AKT (pAKT) in human prostate cancer cells (Chetram et al., 2013). Notably, this ROS-mediated AKT activation is through PTEN, a negative regulator of AKT signaling, because PTEN was rendered catalytically inactive through oxidation by ROS. Importantly, the two vanadium compounds, bis(acetylacetonato)-oxidovanadium(IV) and sodium metavanadate, inhibit cell proliferation via ROS-induced activation of both PI3K/AKT and MAPK/ERK signaling pathways in human pancreatic cancer AsPC-1 cells (Wu et al., 2016). The effects of the two compounds could be counteracted with the antioxidant N-acetylcysteine. Thus, GT-7 could induce ROS thereby increasing both pERK and pAKT.

We also added comments by the reviewer that Overall the conclusion was that pAKT levels affected apoptotic responses to GT-7 in PANC.  in the manuscript.

  1. The reviewer mentioned, “What remains unclear is the actual target of this drug, and therefore its mechanism of action in increasing pERK. How does GT-7 increase pERK is a fundamental and critical question. Does GT-7 increase MEK activity, or reduce phosphatase activity against pERK? These critical questions are not addressed, and maybe beyond the scope of these studies. If so, it leaves this as a rather descriptive study, with observations about the effects of different compounds on pERK, pAKT and apoptosis, but not providing mechanistic insights that would greatly increase impact and significance. We know no more about how GT-7 works after this study than before.”

Re: We thank Reviewer 2 for these pertinent comments. As mentioned by the Reviewer, the question as to how GT-7 increases pERK is beyond the scope of the current manuscript, and we are currently trying to delineate the target proteins of GT-7 by various methods including the identification of the GT-7-binding proteins.

However, we can provide some mechanistic insights into the action and target(s) of this compound by extending the discussions and the responses to comment 4 by the reviewer.

We are currently hypothesizing that GT-7 can stimulate ERK signaling by increasing pERK, either by inducing ROS or serving as an oxidant. This hypothesis is derived from previous publications on the biological activity and the binding protein of ACA, the original compound of ACA-28 and GT-7, as well as a number of reports that many compounds capable of inducing ERK-dependent apoptosis involve ROS induction/generation as a part of the mechanism.

ACA has been reported as an inhibitor for nuclear export of Rev by binding to the Cysteine-529 residue of CRM1, the receptor for NES, thereby inhibiting nuclear export of Rev (Tamura et al., 2009). Tamura et al. reported the formation of the quinone methide intermediate of ACA to be essential for exerting the inhibitory activity of ACA for nuclear export of Rev (Tamura et al., 2010). Furthermore, the treatment of ACA with N-Acetyl- cysteine (NAC) was found to provide the two adducts. As ACA-28 and GT-7 bear a similar structure (two carbonate esters) to ACA regarding the binding to NAC, this prompted us to investigate if ACA-28 (the seed compound of GT-7) forms the quinone methide intermediate as shown below.

We have investigated whether a quinone methide intermediate (F) is generated from ACA-28, as depicted in the following Scheme 1 (see the attached file). As a result, the diastereomeric NAC-adduct (G) was obtained from ACA-28. High-resolution Mass spectrum and HMBC correlation (dashed arrow) supported the structure of the NAC-adducts (G). Thus, the formation of (G) well supported that the intermediate (F) generates from ACA-28, which recapitulated the formation of the intermediate (B) from ACA, suggesting the formation of (G) in the cell. The structure of the quinone methide intermediate (B, F) is extremely unstable and immediately changes to the corresponding d- and z-adducts, such as (C), (D), or (G) in the presence of nucleophiles. Oxygen is also a nucleophile and could convert (F) into the corresponding peroxide (H). Therefore, ACA-28 and GT-7 via the formation of the quinone methide intermediate (F), thereby serving as oxidants to attack the Cysteine residues of the target proteins. We will incorporate the main points of the above descriptions in the Discussion section of the revised manuscript and we have shown these data for your reference only.

Consistent with these chemical reactions, ACA, the original compound has been reported to induce ROS. In ACA- and sodium butyrate-treated cells, intracellular ROS levels, and NADPH oxidase activities were increased in HepG2 human hepatocellular carcinoma cells (Kato et al., 2014). The decrease in cell number after combined treatment of ACA and sodium butyrate was diminished when cells were pretreated with catalase, a strong antioxidant enzyme that breaks down ROS. The authors also showed that ACA, alone or in combination with sodium butyrate, increased pERK. Similar observations of the ACA-mediated induction of ROS and the inhibition of apoptosis induction by a thiol antioxidant NAC have been reported in other cancer cell lines such as NB4 promyelocytic leukemia cells (Ito et al., 2004). We, therefore, assume that GT-7 and ACA-28, similar to the original compound ACA can increase pERK via stimulation of ROS signaling. As mentioned previously, targets of ROS include various phosphatases that inactivate upstream kinases of ERK signaling as well as phosphatases against ERK. Thus, both scenarios remain possible regarding the action of GT-7 in that this agent can increase MEK activity, or reduce phosphatase activity against pERK, depending on the context of the genome or gene profiling expression in each cancer cell line.

  1. The reviewer mentioned, “There is some discussion about the difference in signaling by the various KRAS mutants present in the PANC cell lines (lines 491-505). But, how does this account for the greater sensitivity to GT-7 in the Q61H KRAS cells and relative resistance of the G12D KRAS cells? The association with gene expression described in lines 542-560 is highly speculative and untested by experiments.”

Re: We thank the reviewer for their comments. Regarding the plausible interpretation of the various KRAS mutations and their relevance on the sensitivity to GT-7 would be as follows. If the cells harbor the KRAS G12Dmutation (PANC-1), the mutant KRAS preferably activates AKT signaling, whereas wild-type KRAS activation results in signaling through both ERK and AKT. In contrast, the KRAS Q61H mutation (T3M4) would preferentially activate ERK signaling as compared with AKT. The KRAS G12C mutant displayed relatively low AKT signaling activation. Thus, as the reviewer mentioned in comment 4, the conclusion was that pAKT levels affected apoptotic responses to GT-7 in these PDAC cell lines. Additionally, considering the threshold theory proposed by Park’s lab, it may be that in order to achieve ERK-dependent apoptosis, the higher cellular ERK phosphorylation levels would be more favorable. In this regard, differential gene expression patterns of EREG and ANXA1, which are highly expressed in MIA-Pa-Ca-2 and T3M4 but poorly expressed in PANC-1 might contribute to the more sustained ERK signaling activation preferable to achieve ERK-dependent apoptosis. However, we agree that these discussions on gene expression profiling are speculated based on the previous reports to interpret our observations and need future validation. We, therefore, added the sentences mentioning the limitation of our interpretation.

Reviewer 3 Report

The authors have done a big revision according to the reviewer's suggestions. The revised manuscript is clear and interesting.

Author Response

We are grateful to Reviewer 3 for providing positive responses to our revised manuscript as below. 

              The reviewer made the following observations: “The authors have done a big revision according to the reviewer's suggestions. The revised manuscript is clear and interesting.

Round 2

Reviewer 2 Report

Authors have added several lines of text to address issues raised.

This manuscript is a resubmission of an earlier submission. The following is a list of the peer review reports and author responses from that submission.

Round 1

Reviewer 1 Report

Dear Editor

Please find below the review for the manuscript entitled “ACAGT-007a, an ERK MAPK Signaling Modulator, in Combination with AKT Signaling Inhibition Effectively Induces Apoptosis in Pancreatic Cancer Cells” by Khandakar et al.

This study investigates the effects of the compound GT-7 (ACAGT-007a) on pancreatic cancer cell lines. The authors show that GT-7 promotes increased ERK phosphorylation leading to inhibition of proliferation and to apoptosis. They also investigate the PI3K/AKT signaling pathway that might be required to mediate the GT-7 effect. To look into the mechanism and into the role of signaling components/negative regulators, specific inhibitors are applied. The work reveals new aspects for potential therapeutic approaches for PDAC. The data seem solid, acquired with a quite restricted set of methodologies. However, some experiments and controls need further validation while some claims need additional clarification as delineated in the comments for the authors before being considered for publication. Nevertheless, the work is of interest in its basics and might become suitable after a major (but not extensive) revision.

Figure 1

  • In figure 1, the authors should be careful in interpreting the data at this point, growth inhibition does not necessarily mean cell death/ killing. To better judge proliferation and apoptosis, it would be necessary to also assess how GT-7 affect cell cycle progression and apoptosis at the same time point (48h).
  • 1A and B show results of the same kind of experiment. However, cell viability of all lines is lower in A. For example, with 15 µM GT-7 in MiaPaca, relative cell number is <30% in A, but 40% in B. Same for Panc-1, with 30 µM GT-7, relative cell number is <40% in A, but >50% in B. Please explain or comment on this obvious difference.

Figure 2

  • Please provide relative quantification of cleaved caspase-3.
  • Since induced apoptosis and caspase activation is one of the major points in this study, the authors should better elaborate and demonstrate it by additional means, e.g. Flow cytometry (Annexin V or Caspase staining). Such analysis would allow to quantify apoptosis on the single cell level and, if performed over the whole timeframe (up to 48h), to answer the question whether a selection takes place. Along the same line, a growth curve would be useful to assess whether cells recover after the first hours of treatment when cl. Casp-3 levels decrease at least in T3M4.
  • 2C does not help much. Except for T3M4 0h, the cultures do not look “healthy” as if cells did not attach properly. What is the cell density at the start of the experiment, so 24h after seeding according to the methods? It would be useful to show untreated cells after 10h as a comparison or even a growth curve (up to 48h).

Figure 3

  • Under same conditions as in Fig. 2 (2 h, 30 µM GT-7) pERK and cl. Casp-3 levels seem to be much higher here. Any explanation?

Figure 4

  • Quantification of cl. Casp-3 western blots would be useful to prove the statement that “high AKT phosphorylation levels as exemplified by PANC-1 cells may hamper GT-7-induced apoptosis.” (p. 8, line 307). Based on the intensity of the band it does not look weaker in PANC-1 compared to the other cell lines.
  • As noted in the previous comment to Fig. 2, flow cytometry-based analysis would be more suitable than the images in Fig. 4C to substantiate the findings.

Discussion

  • In the discussion, the authors should give some explanation how or why increased ERK signaling would lead to apoptosis.
  • Differences in the genetic background of the cell lines are mentioned only in the discussion part. Such details should be provided in the beginning of the results.

Reviewer 2 Report

In this study, the authors demonstrated that GT-7, an anti-cancer compound, regulates cell proliferation and apoptosis in MIA-Pa-Ca-2, T3M4 and PANC1, three human PDAC cell lines. Although GT-7 stimulates ERK phosphorylation and exerts antiproliferative effects in PDAC cell lines, the authors state that ERK is not solely responsible for apoptosis induction as evidenced by the failure of the MEK inhibitor to cancel the GT-7-mediated apoptosis. Moreover, the authors show that GT-7 activates both ERK and AKT while downregulates the expression levels of DUSP6 and PTEN, major negative regulators of the ERK and AKT signaling pathways. The PI3K inhibitor Wortmannin enhances GT-7-mediated apoptosis by abolishing AKT activation in MIA-Pa-Ca-2 and T3M4 cells, indicating that AKT activation is a key resistance mechanism against GT-7.

The work is interesting, clear and well conducted. There are some minor points to address:

  1. In Fig.1A the data show that the three lines are differently susceptible to inhibition of proliferation by GT7 at the dose of 30 mM, being MIA-Pa-Ca-2 cell line the most susceptible (18%), T3M4 the intermediate (28%) and PANC1 the least susceptible (38%). The IC50s for the three lines are also reported. In Fig.1B, even if it is confirmed that the MIA-Pa-Ca-2 cell line is the most susceptible and the PANC1 cell line is the least susceptible, the survival rates at GT7 at the dose of 30 mM are different, being MIA-Pa-Ca-2 just above 20%, T3M4 almost 40% and PANC1 slightly above 50%, as well as the IC50s deductible from the histograms are different too. It is not correct to show different amount of proliferation inhibition in the same lines treated in the same way (dose response with GT7 for 48h) in two figures. The authors must standardize the 2 panels by averaging the 6 experiments and give the same survival rates for the inhibition of proliferation and the same IC50s in Fig1A and in Fig1B.
  2. In Fig.2B data of Relative quantification of ERK and AKT phosphorylation levels is shown in setting 100% MIA-Pa-Ca-2 cells, while in Fig.3B and D and in Fig.4B each cell lines were set as 100%. It would be better to uniform the figures 2B with the others.
  3. In Fig.2B, 3B and 4B the authors have to show the Relative quantification of cleaved Caspase-3/full length Caspase-3.
  4. In the discussion the authors declare: “The study demonstrated that GT-7 decreased proliferation and induced apoptosis in PDAC cell lines”. This conclusion is reasonable since data show both inhibition of proliferation and induction of apoptosis in all the three cell lines. However, the susceptibility of the three cell lines to the two events is different. Indeed, while the susceptibility of inhibition of proliferation is stronger in MIA-Pa-Ca-2, intermediate in T3M4 and weaker in PANC1, the susceptibility of induction of apoptosis is stronger in T3M4 than in MIA-Pa-Ca-2. It would be interesting if the authors better discuss this difference giving some possible explanation.

Reviewer 3 Report

Reviewer comments for Cells 1269490

  1. The response to GT-7 (ACAGT-007a) a pharmacological derivative of ACA-28 discovered by screening was tested in three human pancreatic cancer (PDAC) cell lines, MIA-Pa-Ca-2, T3M4 and PANC-1. This study in PDAC replicates similar work in melanoma lines [8], limiting the novelty.
  2. The Introduction emphasized the transforming RAS mutations in PDAC but only in line 410 is it mentioned that T3M4 are not K-RAS mutated, a key point that deserves earlier mention. The cells showed variable sensitivity to GT-7 in the micromolar dose range, with MIA-Pa-Ca-2 more sensitive, and PANC-1 being relatively resistant. These differences do not support unqualified conclusions about the effects of GT-7.
  3. Phospho-ERK was elevated in less than 2 hr after adding 30 uM GT-7 in all three cell lines, and sustained for several hours. An increase in pERK seems contradictory to the reduction in proliferation, for which pERK is often used as a marker. How do the authors account for this? The pERK response cannot be attributed to reduction in DUSP6 shown in Fig. 3, and no alternative explanation is offered.
  4. An increase in pS473 in AKT was transient at 2 hr treatment after GT-7. The parallel AKT responses of the 3 cell lines contrast with the differential sensitivity to GT-7 in terms of cell numbers. This argues against some common mechanism for GT-7 in PDAC.
  5. Cleaved caspase-3 is used as a marker for apoptosis in Fig. 2, but MIA-Pa-Ca-2 and PANC-1 show weak responses to GT-7, less than the T3M4 cells. Images in Fig 2C are small and not especially useful.
  6. 3A shows that GT-7 induction of caspase-3 cleavage was blocked by MEK inhibitor UO126, only in T3M4 cells, not in the other lines. How to account for this? The blots in 3A for caspase-3 and cleaved caspase are unevenly loaded and poor quality.
  7. Based on a previous study by this group using related compound ACA-28, here DUSP6 levels were examined over time +/- GT-7 and +/- MG132 to block proteasome action. This series of experiments is not well connected to the others and offers little about the mechanism of action of GT-7.
  8. Fig 4 examines pS473 in AKT in response to GT-7 +/- Wortmannin and Perifosine. The caspase blots are better than Fig 3, and show Wortmannin enhances the response to 30 uM GT-7 while Perifosine reduces it. Notably, inhibition of PI3K/AKT with these agents did not induce caspase-3 cleavage in PDAC lines.
  9. GT-7 reduced the levels and altered the mobility of PTEN in Westerns in all 3 PDAC lines and the response was not affected by MG132. What was the rationale for examining PTEN levels relative to GT-7?

Overall, the results expose quite different responses among PDAC cell lines (discussed in lines 386-397) that do not support unqualified conclusions such as in lines 377-378 and 60-61. The data presented are a collection of observations about signaling and effects of inhibitors and combinations that do not advance an understanding of the action of GT-7, its potential as a therapeutic agent or identification of a target of GT-7. Just the persistent impressive increase in pERK still lacks a possible hypothesis. The proteasome dependent and independent actions of GT-7 further cloud the issue as to possible mechanisms and one key question (lines 468-469) remains unanswered.